# Impact of patterns of language use and socio-economic status on a constructed response Situational Judgment Test (SJT)

Xuan Pan[1], Vivian Huang[1], Sonia Laumbach[2], H. Liesel Copeland[3], Modupeola Akinola[4], Daryl Rosenbaum[5], Alexander MacIntosh[1]*

1 Department of Research, Acuity Insights, Toronto, Ontario, Canada, 2 Department of Family Medicine, Rutgers Robert Wood Johnson Medical School, Piscataway, New Jersey, United States of America, 3 Office of Admissions, Rutgers Robert Wood Johnson Medical School, Piscataway, New Jersey, United States of America, 4 Department of Pediatrics, Wake Forest University School of Medicine, Winston-Salem, North Carolina, United States of America, 5 Department of Student Affairs and Admissions, Wake Forest University School of Medicine, Winston-Salem, North Carolina, United States of America

* amacintosh@acuityinsights.com

**Data Availability Statement:** There are ethical and legal restrictions pertaining to the release of these potentially personally identifiable data. In summary we are unable to share these data for the following

## Abstract

The study explored the impacts of patterns of language use (PLU) and socio-economic status (SES) on Casper, a constructed-response situational judgment test (SJT). 10,266 applicants from two U.S. medical schools were grouped into self-reported balanced bilinguals, unbalanced bilinguals, English monolinguals, and English as a Second Language (ESL) students. A multicomponent SES composite was used to assess the degree of socioeconomic disadvantage (DSD). Results from a hierarchical regression analysis showed that after accounting for demographic variables, both PLU and DED were significant factors on applicants' Casper performance. Bilingualism was associated with better Casper performance compared to English monolinguals and ESL students. No significant effect of speaking English as a native language was found on applicants' Casper performance. English monolinguals and ESL students performed equivalently on Casper. Finally, high DSD was associated with better Casper performance than low DSD, and the impact of DSD on Casper held the same across all four language groups. These findings provide evidence that socio-cultural factors, such as PLU and DSD have important impacts on SJT performance. Further research is needed to understand the role of differences in language construction across socio-cultural factors on constructed-response SJT performance.

## Introduction

Student diversity has long been a concern in the field of medical education. However, limited progress has been made to address this concern in the United States as indicated by the relative low proportion of medical students who are traditionally underrepresented in medicine (URiM). The Association of American Medical Colleges (AAMC) defines URiM applicants as racial and ethnic populations that are underrepresented in the medical profession relative to their numbers in the general population [1]. However, these factors go beyond race or

reasons: 1 - Third party data - Data from this study are from educational records belonging to the learners (as per FERPA legislation) and cannot be distributed by the researchers as per ethical review. A description of the data set and the third-party source: - The study used data from two medical schools (Rutgers Robert Wood Johnson Medical School and Wake Forest School of Medicine). - Data includes: Applicant identity characteristics (race, socioeconomic status indicators) and educational attainment information (exam and assessment scores) Verification of permission to use the data set: - The study was approved by Wake Forest School of Medicine IRB (IRB00073284) and Rutgers Institutional Review Board (Pro2022001657). The ethical review boards determined that the study meets the criteria for a waiver of HIPAA authorization according to 45 CFR 164.512, accordingly informed written or verbal consent was not obtained. All necessary contact information others would need to apply to gain access to the data: - Wake Forest School of Medicine IRB - irb@wfu.edu - IRB00073284 - Rutgers Institutional Review Board - IRBOffice@research.rutgers.edu - Pro2022001657 2 - Human research data - The data set includes potentially sensitive information that could be used to re-identify an individual based on demographic characteristics and examination dates and scores. As such, ethical restrictions have been set to limit data sharing only to individuals involved in the study. The restrictions in detail: - data contain potentially identifying or sensitive learner information (demographics and test dates and scores) Contact information for a data access committee, ethics committee, or other institutional body to which data requests may be sent: - Wake Forest School of Medicine IRB - irb@wfu.edu - IRB00073284 - Rutgers Institutional Review Board - IRBOffice@research.rutgers.edu - Pro2022001657.

**Funding:** The authors received no specific funding for this work.

**Competing interests:** SL, LC, MA and DR have no disclosures to declare. XP, VH, and AM disclose while writing this article they were salaried employees of Acuity Insights which administers a situational judgment test, Casper. The authors receive no reimbursements, fees, or funding related to this study or its outcomes. This does not alter our adherence to PLOS ONE policies on sharing data and materials.

ethnicity and often include indicators of socioeconomic status and opportunity such as family income, gender identity, rurality, etc. [2]. The performance gap in knowledge-based admission metrics (e.g., grade point average (GPA) and Medical College Admission Test (MCAT)) may have contributed to this discrepancy [3, 4]. Although GPA and MCAT are standard metrics in applicant selection to medical schools given their strong validities for predicting medical school performance [5], both metrics often yield lower performance scores for applicants who are traditionally underrepresented in medicine (URiM) compared to non-URiM applicants [6, 7]. Consequently, this limits the social mandate to widen access to medical education. Moreover, these academic metrics may not be adequate in predicting future performance indicators of non-cognitive characteristics [8] which include social, emotional, and behavioral skills [9]. The need for a holistic admission process that reflects the changing healthcare landscape has been identified in the field of medical education. In recent years, admission processes have aimed to select students who exhibit academic capacity along with the social, emotional, and behavioral skills needed to address complex situations and serve diverse populations.

## Situational Judgment Tests

Over the last decade, Situational Judgment Tests (SJTs) have been increasingly adopted as part of the movement towards more holistic admissions [10, 11]. A body of evidence has supported SJTs' predictive validity [12] and the validity of providing unique information over-and-above cognitive ability and personality tests [1]. Furthermore, demographic disparities have been found to be significantly smaller in SJTs compared to tests that assess cognitive abilities [7, 13]. Simulation studies have also suggested that using SJTs in the admission process has the potential to select more applicants from underrepresented demographic groups [7, 14].

Although SJTs showed smaller demographic effects compared to cognitive tests, the existing SJT literature still identifies differences in SJT performance across race and socio-economic status (SES) with members of the majority typically outperforming those from the minority [14]. In addition, language abilities have also been shown to be associated with performance on SJTs with native English speakers outperforming non-native English speakers [15]. In a recent study with international medical graduates, Patterson and colleagues found that higher English fluency was associated with better SJT performance [16].

To date the above research on demographic disparities have only been completed with fixed-response evaluations. In a typical fixed-response SJT, applicants are presented with social scenarios that they are likely to encounter during future study and employment. They are required to judge the effectiveness of several potential ways to respond to these scenarios [17]. Fixed-response test formats are known to show greater discrepancies between over and underrepresented groups compared to constructed-responses, where applicants can express their arguments and rationales more freely [7, 18, 19]. [20] show that constructed response format SJTs produce smaller minority-majority differences compared to the fixed-response format (d = 0.28 written vs. d = 0.92 multiple choice). Unlike traditional fixed-response SJTs, in constructed-response SJTs, applicants are allowed to describe what they would do in response to the hypothetical scenario and why [21–24]. As such, constructed-response SJTs allow applicants to describe life experiences when responding to scenarios. This may unintentionally disadvantage non-native English speakers, whose word choice and writing structure may differ from that of native English speakers [16, 25]. However, it is also possible that non-native English speakers and bilinguals have different life experiences to draw from, thus giving them unique perspectives in constructed-response SJTs that are different from native speakers.

In addition to language, applicants' SES has also been shown to be associated with their SJT performance with applicants from high SES outperforming those from low SES [7, 26].

Moreover, data from the 2019 U.S. Census Bureau indicates households with limited English-speaking ability disproportionately represent lower income families. Census data showed 17% of households not in poverty had limited English-speaking compared to 36% of households in poverty [27]. Further, bilingualism could also be associated with high SES and access to high quality educational resources and opportunities. Previous studies have found that students with high SES demonstrated a more positive attitude and motivation towards learning a second language [28], and a strong positive correlation between SES and achievement in learning a second language [29, 30]. Therefore, it is possible that language and SES may be intertwined to impact performance on constructed-response SJTs.

To our knowledge, no studies have examined the relationships between patterns of language use (PLU) and SES on performance in constructed-response SJTs used for high-stakes admissions selection. Understanding these associations is critical both practically and theoretically. Practically, admission committees employing these selection measures require an understanding of the degree to which individuals from diverse backgrounds can effectively respond. Theoretically, understanding how PLU, SES and the constructed-response evaluation structure interact can clarify how an individuals' socio-cultural upbringing can be expressed through and perceived by SJT evaluations.

The aim of this study was to examine the impacts of PLU and SES on performance in a constructed-response SJT assessing professionalism for entry to medical school. We hypothesize that there will be main effect differences in SJT performance by both PLU and SES. Specifically, that English monolinguals, bilinguals whose first language is English (Unbalanced Bilinguals) and those with higher SES will perform better compared to ESL students and those with lower SES.

## Method

### Data structure

Application data from the 2019–2020 admissions cycle of two medical schools located in the United States were included in this retrospective analysis. The study was approved by Wake Forest School of Medicine IRB (IRB00073284) and Rutgers Institutional Review Board (Pro2022001657). The ethical review boards determined that the study meets the criteria for a waiver of HIPAA authorization according to 45 CFR 164.512, and therefore informed written or verbal consent was not obtained. SES indicator data were extracted from the American Medical College Application Service (AMCAS) system for 10,266 applicants (20% of all medical school applicants). These data include education and occupation (EO) indicator for an applicant's parent(s), family income, Federal Pell Grant recipient, AAMC Fee Assistance Program (FAP) recipient, and family receives financial aid during childhood status. The AAMC's EO indicator has five levels, however the first two levels functionally identify applicants as coming from a disadvantaged background. This designation is based on a combination of the parent's education and nature of employment. Applicants whose parents have 'less than a bachelor's degree and/ or work in 'Service, clerical, skilled, and unskilled labor' are designated as EO 1/2 [31].

Race and language use data were obtained from a voluntary demographic survey offered to applicants after writing the Casper SJT, which all applicants completed. Casper is a constructed-response SJT intended to assess social intelligence and professionalism. Applicants were presented with 12 text or video scenarios designed to probe for social, emotional, and behavioral aspects such as empathy, communication, and resilience among others [18]. Each scenario is followed by three open-ended behavioral tendency questions. Applicants have five minutes to type their responses. Each scenario is scored by a different trained human rater on

**Table 1. Patterns of Language Use (PLU) groups.**

| PLU | Description |
| --- | --- |
| English monolingual | Never learned a language other than English |
| Unbalanced bilingual | Speaking a second language other than English but always/often spoke English at home and rarely/from time to time spoke another language at home |
| Balanced bilingual | Always/often speak English and always/often speak another language at home |
| ESL student | Often/always speak another language at home but English only from time to time or rarely/never at home |

*Note*. Applicants were grouped into four PLU groups based on their self-reported languages used and frequencies of language use at home.

a norm-referenced scale. Raters use a 1 to 9 Likert scale and an applicant's final score is the average of their 12-section scores, standardized against other testers in their cohort.

PLU were determined based on applicants' self-disclosed frequency of using English at home and frequency of using a language other than English at home. Applicants were grouped into four PLU groups: 1) English monolingual, 2) Unbalanced bilingual who mainly use English at home, 3) Balanced bilingual who frequently use two languages at home, 4) ESL student (see Table 1 for detailed descriptions). Race groups were determined based on the self-disclosed race information that applicants provided. Applicants were grouped into five race groups: 1) White, 2) Black/African-American, 3) Hispanic, 4) Asian, 5) Other. See Table 2 for a summary of the demographics of the study population.

## Data analysis

First, to implement a more sensitive and reliable measure of SES [32], a composite measure of SES was calculated based on work by [33, 34]. This was done to allow SES to be presented as a continuous indicator–Degree of Socioeconomic Disadvantage (DSD). To build the DSD indicator, the following five variables were used to predict applicants' AAMC SES status (education and occupation (EO) indicator for an applicant's parent(s): EO1/2 vs EO3/4/5) in a logistic regression: 1) gross family income, 2) highest parental education, 3) Federal Pell Grant recipient, 4) AAMC Fee Assistance Program (FAP) recipient, 5) family receives financial aid during childhood. Predicted probabilities of being identified as EO1/2 (the SES disadvantage status in the AAMC application) were calculated based on the regression model. To evaluate the fit of the DSD indicator, the distinguishing ability of the model was evaluated using the Area under the Receiver Operator Characteristic (ROC) Curve (AUC). With acceptable distinguishing ability, DSD was used in the next step of the analysis to address the primary question.

To explore the relationship between PLU and DSD and their associations to Casper performance, a linear regression was conducted with PLU used as the predictor and DSD as the dependent variable. Next, a hierarchical linear regression analysis was used to address how PLU and DSD are associated with Casper performance. Four variables (medical school applied, along with age, gender, race demographics) were included in the regression model as covariates in step one. These demographic variables have been found to influence standardized test performance including SJTs [7, 13, 14, 35–37]. PLU entered the model in step two, DSD entered the model in step three, and the interaction between PLU and DSD entered the model in step four. Casper score was used as the dependent variable. All analyses were done in R (version 4.1.2). The significance of the predictors was determined with the type-II Wald tests using the Anova function provided by the car package (version 3.0–11) [38]. Model comparisons

**Table 2. Demographics.**

| Demographic Measure | Study Population |
| --- | --- |
| Age | Mean = 24.4 |
| | SD = 2.56 |
| | Range: 18–67 |
| Gender | Female: 5417 (52.8%) |
| | Male: 4849 (47.2%) |
| Race | White: 4955 (48.3%) |
| | Asian: 3120 (30.4%) |
| | Black/African-American: 640 (6.2%) |
| | Hispanic: 788 (7.7%) |
| | Other: 720 (7.0%) |
| Patterns of Language Use | English monolingual: 2304 (22.4%) |
| | Unbalanced bilingual: 4030 (39.3%) |
| | Balanced bilingual: 2939 (28.6%) |
| | ESL student: 993 (9.7%) |
| Gross Family Income | Less than $25,000: 557 (5.4%) |
| | $25,000 to $50,000: 1045 (10.2%) |
| | $50,000 to $75,000: 1169 (11.4%) |
| | More than $75,000: 7495 (73%) |
| Parental Highest Education | Less than high school: 197 (1.9%) |
| | High school graduate: 612 (6.0%) |
| | Some College: 349 (3.4%) |
| | College Graduate: 2778 (27.1%) |
| | Graduate and/or professional degree: 6330 (61.7%) |
| Fee Assistance Program Recipient (FAP) | Yes: 1149 (11.2%) |
| | No: 9117 (88.8%) |
| Family receiving financial aid during childhood | Yes: 2396 (23.3%) |
| | No: 7870 (76.7%) |
| Pell Grant Recipient | Yes: 2310 (22.5%) |
| | No: 7956 (77.5%) |
| AAMC SES Indicator (EO Status) | EO1/2: 1991 (19.4%) |
| | EO3/4/5: 8275 (80.6%) |

*Note*. Demographic data obtained from the AMCAS applicant system. All demographic questions included a "Decline to Answer" option, which was converted to NA and excluded from the data analyses.

were done with likelihood ratio tests using the lrtest function from the lmtest package (version 0.9–39) [39].

## Results

### Constructing DSD from SES

The five SES related variables together were a significant predictor of applicants' EO status ($R^2$ = 0.375) (see Table 3 for full model outputs). The predictive accuracy was acceptable at 87% (AUC = 0.87). Therefore, this linear indicator of SES, the DSD indicator was suitable towards addressing the study aim. Predicted probabilities of being identified as EO1/2 (SES disadvantage) were calculated and used as the degree of socioeconomic disadvantage (DSD) in the following analyses.

**Table 3. Model output for the composite measure of the degree of socioeconomic disadvantage (DSD).**

| Coefficients | Estimates | Std. Error | Z value | p value |
|---|---|---|---|---|
| Intercept | -0.96 | 0.11 | -8.29 | < .001*** |
| Pell Grant: Yes | 0.68 | 0.09 | 7.64 | < .001*** |
| Fee Assistance Program (FAP): Yes | 0.22 | 0.10 | 2.26 | .02 |
| Gross Family Income: $50,000 to $75,000 | 0.08 | 0.11 | 0.77 | .44 |
| Gross Family Income: Less than $25,000 | -0.50 | 0.13 | -3.80 | < .001*** |
| Gross Family Income: More than $75,000 | -0.63 | 0.11 | -5.64 | < .001*** |
| Parental Highest Education: Graduate and/or Professional Degree | -1.63 | 0.07 | -22.16 | < .001*** |
| Parental Highest Education: High School Graduate | 1.82 | 0.11 | 15.90 | < .001*** |
| Parental Highest Education: Less than High School | 1.10 | 0.17 | 6.38 | < .001*** |
| Parental Highest Education: Some College | 2.30 | 0.15 | 15.15 | < .001*** |
| Family Receiving Financial Aid During Childhood: Yes | 0.30 | 0.08 | 3.83 | < .001*** |
| $R^2$ Tjur = 0.375 | | | | |

*Note.* Degree of socioeconomic disadvantage was calculated from the probabilities of being identified as EO1/2 based on the regression model with five SES-related predictors: 1) gross family income, 2) highest parental education, 3) whether to receive Pell Grant, 4) whether to receive FAP, 5) whether family receive financial aid during childhood.

## Relationship between PLU and DSD

PLU was a significant predictor of DSD ($F(3) = 250.1$, $p < .001$). ESL students had significantly a higher probability of being in a lower DSD than English monolinguals ($\beta = 0.21$, $p < .001$). At the same time, the DSD of English monolinguals, unbalanced bilinguals ($\beta = -0.02$), and balanced bilinguals ($\beta = 0.03$) were similar. See Table 4 for mean DSD for each PLU group.

## The impacts of PLU and DSD on Casper performance

The four covariates (school applied, age, gender, race) added significantly to the model's explanatory power ($p < .001$), thus they were retained in the model to address the main question. Regarding PLU, balanced bilinguals ($\beta = 0.13$, $p < .001$) and unbalanced bilinguals ($\beta = 0.14$, $p < .001$) scored higher on Casper when compared to English monolinguals, while ESL students performed similarly to English monolinguals ($\beta = -0.006$, $p = .86$). Model comparisons showed that adding PLU significantly improved the model's explanatory power ($\chi^2(2) = 69.86$, $p < .001$). DSD showed a significant impact on Casper performance with greater advantage being associated with higher Casper scores ($\beta = -0.30$, $p < .001$). Model comparisons showed that adding DSD significantly improved the model's explanatory power ($\chi^2(1) = 61.22$,

**Table 4. Mean degrees of socioeconomic disadvantage (DSD) for Patterns of Language Use (PLU) groups (standard deviations in brackets).**

| Patterns of Language Use (PLU) | Mean degree of socioeconomic disadvantage (DSD) |
|---|---|
| English monolingual | 0.18 (0.22) |
| Unbalanced bilingual | 0.15 (0.20) |
| Balanced bilingual | 0.20 (0.25) |
| ESL student | 0.38 (0.32) |

*Note.* Applicants' PLU and DSD are correlated. English monolinguals (mean DSD = 0.18), unbalanced bilinguals (0.15), and balanced bilinguals (0.20) are less likely to have a lower DSD than ESL students (0.38).

$p < .001$). Finally, the interaction between PLU and DSD did not reach significance ($\chi^2(3) =$ 7.58, $p = .06$). Together, PLU and DSD explained 1.20% of the variance in applicants' Casper scores. The relative contributions of the predictors and $R^2$ changes between models can be found in Table 5.

## Discussion

The current study investigated the impacts of PLU and SES on the performance in a con-structed-response SJT, Casper, after accounting for demographic variables among about 20% of applicants to all U.S. medical schools. PLU was found to be associated with Casper perfor-mance. Unbalanced bilinguals whose first language was English and balanced bilinguals who used two languages frequently at home had statistically better Casper scores than English monolinguals and ESL students who did not use English frequently at home. Interestingly, English monolinguals and ESL students had similar Casper scores.

Bilingualism has long been suggested as a method of maintaining brain structure and pre-venting cognitive decline, resulting in cognitive advantages including advanced cognitive switching and inhibition abilities [40]. According to [41], the "constant strain of language management on the conflict-monitoring system" and extra effort required to frequently inhibit one language and switch between languages leads to enhanced neural pathways and stronger executive control abilities, especially inhibition and cognitive switching. The advanced

**Table 5. Model outputs for the hierarchical regression analysis.**

| Coefficients | Estimates | | | | |
|---|---|---|---|---|---|
| | Intercept Only | School Applied + Age + Gender + Race | PLU | DSD | PLU x DSD |
| Intercept | 0.31*** | 1.17*** | 1.02*** | 0.97*** | 0.96*** |
| School Applied: 1 | | 0.07*** | 0.08*** | 0.08*** | 0.08*** |
| School Applied: Both | | 0.24** | 0.24** | 0.24** | 0.24** |
| Age | | -0.03*** | -0.03*** | -0.03*** | -0.03*** |
| Gender: Male | | -0.18*** | -0.18*** | -0.17*** | -0.17*** |
| Race: Asian | | 0.07** | 0.06* | 0.05* | 0.05* |
| Race: Other | | 0.03 | 0.02 | 0.03 | 0.03 |
| Race: Hispanic | | -0.16*** | -0.17*** | -0.14*** | -0.14*** |
| Race: Black/African-American | | -0.53*** | -0.53*** | -0.49*** | -0.49*** |
| PLU: Unbalanced Bilingual | | | 0.14*** | 0.14*** | 0.14*** |
| PLU: ESL student | | | -0.06 | -0.00 | -0.03 |
| PLU: Balanced Bilingual | | | 0.14*** | 0.15*** | 0.19*** |
| DSD | | | | -0.31*** | -0.24** |
| PLU: Unbalanced Bilingual x DSD | | | | | 0.00 |
| PLU: ESL student x DSD | | | | | 0.03 |
| PLU: Balanced Bilingual x DSD | | | | | -0.21* |
| $R^2$ | 0.000 | 0.047 | 0.053 | 0.059 | 0.060 |

Note.

* $p < .05$,

** $p < .01$,

*** $p < .001$.

The reference level for each of the categorical variables was: Female, White, and English monolingual. Demographic variables (school applied, age, gender, race) explained 4.7% of the variance in applicants' Casper scores. Both PLU and DSD added little but significantly to the model's explanatory power; together they explained 1.2% of the variance in applicants' Casper scores. No significant interaction between PLU and DSD was found.

inhibition and switching abilities could in turn benefit applicants' performance on SJTs. For example, better cognitive switching ability could lead to flexibility in thinking, thus making it easier to discover novel perspectives and solutions for social dilemmas.

Consistent with previous work of other assessments, the current study found an association between SES and Casper scores [42]. And like [43], these differences were smaller in the SJT than what is typically observed with academic evaluations. This gap is thought to exist for several reasons. First, individuals with lower SES generally have had less access to opportunities due to financial and social resource constraints affecting educational and experiential opportunities including mentorship, volunteering, and extra-curriculars to name a few [44, 45]. Compounding this opportunity gap are the strained mental and emotional experiences many who live with lower SES have [46]. These psychological barriers include being more likely to feel greater emotional distress, challenging feelings of belonging and imposter syndrome in college, and less perceived support from family. All of which may contribute to the trend in lower educational attainment [46]. Lastly, the structure common in U.S. higher education favour people from higher SES backgrounds. The practices, behaviours, and cultural ideals of higher SES households are centered in higher education, leaving people from families with lower SES othered and in unfamiliar territory [47, 48]. While there is no real basis for any SES group difference in capacity or aptitude of situational judgment, the gaps in opportunity, elevated psychological barriers, and centering of higher-SES values have real effects and requires improvement. Future work should investigate a wide range of SES proxies and how their associated experiences have impacted applicants [49]. These learnings can be used to inform the selection process, modify educational values, and improve evaluation measures.

Another possible reason that both balanced and unbalanced bilinguals performed better than ESL students may be that bilingualism is associated with higher SES and access to high quality educational resources and opportunities. Consistent with previous studies suggesting that higher SES is associated with more positive attitude, motivation, and higher achievement in second language learning [28–30], the current results showed that bilinguals (mean DSD = 0.17) are less likely to be in a lower SES than ESL students (0.38). The opportunity to learn another language therefore could be one of many mechanisms by which individuals with higher SES show better performance in SJTs. Future work towards understanding this can include evaluating the mediating effect of language use (e.g., word choice and writing style) on the relationship between SES and SJT performance.

Different from previous findings on fixed-response SJTs where native English speakers outperformed non-native English speakers [15], the current results showed that English monolinguals and ESL students performed similarly on Casper. This, combined with the observation that no significant interaction between PLU and SES was found suggests that being a nonnative English speaker and not regularly using English at home does not inherently place applicants at a disadvantage in the constructed-response SJT for medical school applications. This provides further evidence that constructed-response test formats tend to show smaller discrepancies compared to fixed-response evaluations.

The current paper supports previous findings that SJT performance has smaller differences in performance between advantaged and disadvantaged groups [43]. While supportive in widening access compared to cognitive-focused evaluations, the remaining differential attainment gap may be related to additional barriers faced by individuals from lower SES backgrounds. [44] suggest that access to more information about the admissions process through their parents or social networks may positively impact test scores for advantaged individuals. Further, [26] highlighted that the additional opportunities advantaged individuals have through support from their home and redu ced financial barriers can impact admissions test scores as well. Together, these findings along with previous work emphasize the need to look at socio-

cultural factors, such as language and SES, towards better understanding their combined impact on SJT performance, especially constructed-response SJTs [7, 14, 26, 43]. For instance, some authors [26] have speculated on a variety of variables that might interact with SES to impact SJT performance, such as gender and race. The current findings showed that PLU and SES do not interact in their influences on Casper performance. Future research should expand our work to other constructed-response SJTs and further examine the effect of other socio-cultural factors on SJT performance.

Consistent with previous findings [7, 19], Casper performance was associated with the demographic control variables. Gender, and race in particular have been widely evaluated in admissions. As observed in the current study, a review by [16] of SJTs in medical education and training show that females tend to perform better than males. Similarly, individuals from the majority racial and ethnic group tend to have higher scores than those of the minority.

These results have mixed implications. Interviews, such as multiple mini-interview (MMI), rely on language proficiency and communication skills within a single language. Studies had shown that MMI score was associated with English language proficiency [50] and students whose first language was not English achieved significantly lower scores on MMI than their English-speaking counterparts [51]. Current findings that non-native English speakers were not inherently placed at a disadvantage in the constructed-response SJT may have great implications for medical admissions to increase diversity from a language-cultural perspective. However, the impact of SES on MMI performance are mixed [34]. The current study highlights the importance of considering the interaction of PLU and SES in medical admissions. It is possible that bilingual applicants increase diversity from a language-cultural perspective but not in a socio-economic perspective. More research is needed to ensure that admissions processes take advantage of the skills that bilingual applicants offer and not inadvertently disadvantage applicants without the same socio-economic opportunities.

A salient limitation of our study is that we focused on only one aspect of language use and grouped applicants based on their native languages and frequencies of language use at home. Other language and socio-cultural factors, such as English proficiency or exposure to Western culture are not addressed in the current study. Future research could separate language use and English proficiency to further examine the impact of PLU on SJT performance. Second, the current study only examined applicants to two U.S. medical schools; thus, findings may not be generalizable to the entire AMCAS cohort. These findings may also differ by geography. For instance, in Canada and Australia where Casper is also used there are different patterns in the distribution of race, degree of socioeconomic disadvantage, and patterns of language use. For instance, [52] found similar trends in Casper performance across many demographic groups, however French language use and proficiency was an area of significant investigation in the Canadian context. As such, the origin of these individuals and how that interacts with the region in which they apply are important areas of future investigation. In addition, while sample size and variance across groups support the application of the model, it is possible that the balanced representation of the U.S. population in this study incompletely reflects the relationship between Casper performance and sociodemographic factors. Unfortunately, these distributions are reflective of the medical school applicant community. For instance, 20% of applicants were identified as disadvantaged by AAMC EO status and 2% of applicants indicated having a parent with less than high school degree. Therefore, these findings are more likely to be directly applicable fields where similar demographic profiles are observed in the applicant pool. Third, our study is retrospective and there could be potential confounding factors that were not recorded in the data, thus not accounted for in the current study. Both PLU and SES were determined based on self-reported data, which could be subjective to social desirability bias and could be low in accuracy [53]. It would be beneficial to use longitudinal

designs with repeated measurements to draw further insights, for instance on the developmental trajectories of the impact of learning a second language on SJT performance.

## Conclusion

This study showed that when controlling for demographic variables (age, gender, race), patterns of language use have a small but significant impact on applicants' Casper performance. Knowing a second language for English native speakers significantly increases their Casper performance, potentially through enhanced cognitive flexibility and cognitive control abilities. Interestingly, English monolinguals and ESL students performed similarly on Casper. The interrelated mechanism of this impact is not yet completely understood. Consistent with previous work, higher SES related to higher SJT performance, but the gap was smaller than differences observed with cognitive-focused tests. Admissions policy makers can use this to inform their recruitment processes by increasing resources to disadvantaged areas where applicants have limited access to information and opportunity. Tool designers can improve tests by constructing the assessment to incorporate and value the barriers disadvantaged applicants face potentially through understanding how these barriers are expressed in the applicants' free-text responses. Research is currently underway to investigate differences in writing style and linguistic features as a function of PLU and SES. Our future research is therefore expanding the current work by focusing on why there's an impact of pattern of language use and SES on Casper performance.

## Acknowledgments

We thank Rodica Ivan, Jordan L. Ho, and Sara M. Hejri for their feedback on an earlier version of this work.

## Author Contributions

**Conceptualization:** Xuan Pan, Vivian Huang, Sonia Laumbach, H. Liesel Copeland, Modupeola Akinola, Daryl Rosenbaum, Alexander MacIntosh.

**Data curation:** Xuan Pan, Vivian Huang, Sonia Laumbach, H. Liesel Copeland, Modupeola Akinola, Daryl Rosenbaum, Alexander MacIntosh.

**Formal analysis:** Xuan Pan, Alexander MacIntosh.

**Investigation:** Xuan Pan, Alexander MacIntosh.

**Methodology:** Xuan Pan, Vivian Huang, Alexander MacIntosh.

**Project administration:** Xuan Pan, Sonia Laumbach, H. Liesel Copeland, Modupeola Akinola, Daryl Rosenbaum, Alexander MacIntosh.

**Supervision:** Sonia Laumbach, H. Liesel Copeland, Modupeola Akinola, Daryl Rosenbaum, Alexander MacIntosh.

**Validation:** Xuan Pan, Sonia Laumbach, H. Liesel Copeland, Modupeola Akinola, Daryl Rosenbaum, Alexander MacIntosh.

**Visualization:** Xuan Pan, Vivian Huang.

**Writing – original draft:** Xuan Pan, Vivian Huang.

**Writing – review & editing:** Xuan Pan, Alexander MacIntosh.

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
