## [Decision Letter · Decision Letter 0]

31 May 2023

PONE-D-23-08475Impact of patterns of language use and socio-economic status on a constructed response Situational Judgment Test (SJT)PLOS ONE

Dear Dr. MacIntosh,

Thank you for submitting your manuscript to PLOS ONE. After careful consideration, we feel that it has merit but does not fully meet PLOS ONE’s publication criteria as it currently stands. Therefore, we invite you to submit a revised version of the manuscript that addresses the points raised during the review process.

Please check the reviewers' comments and act accordingly. 

We look forward to receiving your revised manuscript.

Kind regards,

Anastassia Zabrodskaja, Ph.D.

Academic Editor

PLOS ONE

Journal Requirements:

“SL, LC, MA and DR have no disclosures to declare. XP, VH, and AM disclose

while writing this article they were salaried employees of Acuity Insights which administers a situational judgment test, Casper. The authors receive no reimbursements, fees, or funding related to this study or its outcomes.”

Reviewers' comments:

Reviewer's Responses to Questions

**Comments to the Author**

1. Is the manuscript technically sound, and do the data support the conclusions?

Reviewer #1: Yes

Reviewer #2: Partly

2. Has the statistical analysis been performed appropriately and rigorously? 

Reviewer #1: Yes

Reviewer #2: I Don't Know

3. Have the authors made all data underlying the findings in their manuscript fully available?

Reviewer #1: No

Reviewer #2: No

4. Is the manuscript presented in an intelligible fashion and written in standard English?

Reviewer #1: Yes

Reviewer #2: Yes

5. Review Comments to the Author

Reviewer #1: Thank you for the opportunity to review your manuscript. It was well written, included a logical flow, and was very useful for those who utilize SJTs. I have no comments because I truly did not find any concerns with the information presented. The only consideration may be details about the statistical model for greater clarity for those who are not as familiar with the statistics, but it is not necessary.

Reviewer #2: Is the manuscript technically sound, and do the data support the conclusions?

Response: Partly

• The research contains a large sample size that has the power to yield meaningful results, however, differences in size of groups should be recognised.

• Authors acknowledge the limitation in their sample being US only. It would be beneficial to acknowledge the implications of this on the generalizability of the results to different cultures and contexts that use Casper.

• The inclusion criteria for SES is quite rigorous (based on five different indicators (gross family income, parental highest education, fee assistance program recipient, family receiving financial aid during childhood, Pell grant recipient)) and the categories used for both patterns of language use (PLU) are logical. However, they argue that candidates with ESL (English second language) are more likely to be immigrants which is associated with lower SES which is a questionable argument. I would also argue that the research referenced (28, 29) does not clearly justify the claim.

• The control variables (race, gender, school applied) are appropriate to the experiment.

• Overall, I think the data generally does support the conclusions that bilingualism was associated with better Casper performance compared to English monolinguals and ESL students. However, there is a lack of explanation for the finding that high SES was associated with better Casper performance than low SES. I would encourage the authors to suggest why this might be the case? The discussion needs to be strengthened in its commentary about the findings in general and possible underlying caudal factors, alongside suggestions for future research.

Has the statistical analysis been performed appropriately and rigorously?

Response: I don’t know.

• The use of a hierarchical linear regression makes sense to see how much of the variance in Casper scores can be explained by the different predictors. The way in which variables were imputed into the hierarchical model is logical (demographic confounding variables first, followed by the predictor variables in separate steps followed by the interactions), allowing for an understanding of the amount of variance explained in Casper scores by each of the predictors separately while controlling for confounding variables.

• I also believe there is an error in the note in table 5. It says that the variance explained by the demographic variables is 46%. This is not suggested within the table.

• I am a little confused by the numbers that have been reported. Why is explanatory power reported? R2 and R2 change is not appropriately considered. The explanatory power reported looked like a multinomial logistic regression? Could they be confusing the 2? Is it just a post hoc calculation? This needs clarification.

Have the authors made all data underlying the findings in their manuscript fully available?

Response: No

• I don’t see any appendices where the full data is available for us to have a look at.

Is the manuscript presented in an intelligible fashion and written in standard English?

Response: Yes

• The language is simple, easy to follow and appropriate. Overall, the article flows well and I consider it well written. It is a bit difficult to follow when the authors go from referring to SES to DSD, it would be beneficial for this section to be made clearer.

The aims of the study are clear and original. These findings have the potential to inform future test design and also inform decisions of policy makers who may be weighing up using alternative types of tests. Therefore, I agree this research is a valuable addition to the literature in this area and as such, recommend revise and resubmit.

6. PLOS authors have the option to publish the peer review history of their article (what does this mean?). If published, this will include your full peer review and any attached files.

Reviewer #1: No

Reviewer #2: No

---

## [Author Response · Author response to Decision Letter 0]

10 Jul 2023

Please see the 'Response to Reviewer.docx' for the full itemized list of responses.

---

## [Decision Letter · Decision Letter 1]

19 Jul 2023

Impact of patterns of language use and socio-economic status on a constructed response Situational Judgment Test (SJT)

PONE-D-23-08475R1

Dear Dr. MacIntosh,

We’re pleased to inform you that your manuscript has been judged scientifically suitable for publication and will be formally accepted for publication once it meets all outstanding technical requirements.

Kind regards,

Anastassia Zabrodskaja, Ph.D.

Academic Editor

PLOS ONE

Additional Editor Comments (optional):

Reviewers' comments:

Reviewer's Responses to Questions

**Comments to the Author**

1. If the authors have adequately addressed your comments raised in a previous round of review and you feel that this manuscript is now acceptable for publication, you may indicate that here to bypass the “Comments to the Author” section, enter your conflict of interest statement in the “Confidential to Editor” section, and submit your "Accept" recommendation.

Reviewer #1: All comments have been addressed

Reviewer #2: All comments have been addressed

2. Is the manuscript technically sound, and do the data support the conclusions?

Reviewer #1: Yes

Reviewer #2: Yes

3. Has the statistical analysis been performed appropriately and rigorously? 

Reviewer #1: Yes

Reviewer #2: N/A

4. Have the authors made all data underlying the findings in their manuscript fully available?

Reviewer #1: Yes

Reviewer #2: Yes

5. Is the manuscript presented in an intelligible fashion and written in standard English?

Reviewer #1: Yes

Reviewer #2: Yes

6. Review Comments to the Author

Reviewer #1: Thank you for addressing the requests.

Reviewer #2: The revisions made in this revised version of this paper have addressed my comments satisfactorily in my original review. The authors have now amended the wording where we suggested, added clarifications and explanation, and corrected the couple of errors picked up.

I do not require further revisions and so would recommend accept for publication.

7. PLOS authors have the option to publish the peer review history of their article (what does this mean?). If published, this will include your full peer review and any attached files.

Reviewer #1: No

Reviewer #2: No

---

## [Editor Report · Acceptance letter]

24 Jul 2023

PONE-D-23-08475R1 

Impact of patterns of language use and socio-economic status on a constructed response Situational Judgment Test (SJT) 

Dear Dr. MacIntosh:

I'm pleased to inform you that your manuscript has been deemed suitable for publication in PLOS ONE. Congratulations! Your manuscript is now with our production department. 

Kind regards, 

on behalf of

Professor Anastassia Zabrodskaja 

Academic Editor

PLOS ONE